**Measurement report:**
**Intra-annual Variability of Black/Brown Carbon and Its Interrelation with**
**Meteorological Conditions over Gangtok, Sikkim**
Pramod Kumar[1], Khushboo Sharma[1], Ankita Malu[2], Rajeev Rajak[2], Aparna Gupta[1],
Bidyutjyoti Baruah[1], Shailesh Yadav[1], Thupstan Angchuk[1], Jayant Sharma[1], Anil Kumar
Misra[1], Nishchal Wanjari[1], and Rakesh Kumar Ranjan[1#]
[1]DST's Centre of excellence on Water Resources, Cryosphere and Climate Change Studies,
Department of Geology, Sikkim University, Gangtok, Sikkim, India -737102
[2]Department of Geology, Sikkim University, Gangtok, Sikkim, India -737102
[#]Corresponding Author: rkranjan@cus.ac.in
**Abstract**
Black carbon (BC) and brown carbon (BrC) have versatile natures, and they have an apparent
role in climate variability and changes. As the anthropogenic activity is surging, the BC and
BrC are also reportedly increasing. So, the monitoring of BC/BrC and observation of land use
land cover changes (LULCC) at a regional level are necessary for the various interconnected
meteorological phenomena changes. The current study investigates BC, BrC, $CO_2$, BC from
fossil fuels ($BC_{ff}$), BC from biomass burning ($BC_{bb}$), LULCC, and their relationship to the
corresponding meteorological conditions over Gangtok in the Sikkim Himalayan region. The
concentration of BC (BrC) 43.5 µg/m$^3$ (32.0 µg/m$^3$) was found to be highest during the March-
2022 (April-2021). Surface pressure exhibits a significant positive correlation with BC, $BC_{ff}$,
$BC_{bb}$, and BrC. Higher surface pressure results in a calmer and more stable boundary layer,
which effectively retains deposited contaminants. Conversely, the wind appears to facilitate
the dispersion of pollutants, showing a strong negative correlation. The fact that all pollutants
and precipitation have been shown to behave similarly points to moist scavenging of the
pollutants. Despite the dense cloud cover, it is clear that the area is not receiving convective
precipitation, implying that orographic precipitation is occurring over the region. Most of
Sikkim receives convective rain from May to September, indicating that the region has
significant convective activity contributed from the Bay of Bengal during the monsoon season.
Furthermore, monsoon months have the lowest concentrations of BC, $BC_{bb}$, $BC_{ff}$, and BrC,
suggesting the potential of convective rain (as rain out scavenging) to remove most of the
pollutants.
*Keywords:* Black carbon; Brown carbon; LULC; Sikkim Himalaya; Meteorology; Biomass
burning; Radiative forcing.

## 1.0 Introduction

Black carbon (BC), and brown carbon (BrC), are part of fine particulates in air pollution that have a deceptive role in climate variability and changes. BC/BrC is a short-lived climate pollutant with a lifetime of only days to weeks after release in the atmosphere (Pierrehumbert, 2014). During this short period of time, BC/BrC can have significant direct and indirect impacts on the climate, cryosphere, agriculture, and human health (Shindell et al., 2012). It consists of pure carbon in several interconnected forms. BC is formed through the incomplete combustion of fossil fuels, biofuel, and biomass, and is one of the main types of particles in both anthropogenic and naturally occurring soot (Bond et al., 2004). BrC in the atmosphere has been attributed to the burning of biomass and fossil fuels, the biogenic release of fungi, plant debris, and humic matter, and multiphase reactions between the gas-phase, particulate, and cloud microdroplet constituents in the atmosphere (Laskin et al., 2015). BC/BrC is transported from its source to many locations across the world (Ramanathan and Carmichael, 2008). The BC/BrC released into the atmosphere exhibits vertical distribution and follows the prevailing wind speed and direction. It engages with various atmospheric components before eventually settling on the Earth's surface through either wet or dry deposition processes. Its hygroscopic properties render it more prone to cloud seeding and cloud formation, thereby contributing directly to the precipitation mechanism in regions with high humidity (Stevens and Feingold, 2009). In addition, it absorbs both incoming and outgoing radiation, atmospheric BC/BrC modifies radiative forcing, disturbs atmospheric stability, regional circulation, and rainfall pattern, affects cloud albedo, material damage, reduces agricultural productivity, degrades ecosystem, and affects human health (Zhang et al., 2013). However, due to an insufficiency of observations, BrC is one of the least understood and uncertain warming agents (Yue et al., 2022). Numerous studies have been conducted to analyze the global distribution of BC and BrC, including research focused on these species within India as well (Reddy and Venkataraman, 2002a, 2002b; Venkataraman et al., 2006; Park et al., 2010; Sloss, 2012; Helin et al., 2021; 2020; Kumar et al., 2020a; Watham et al., 2021; Bhat et al., 2022; Runa et al., 2022; Yue et al., 2022; Kumar et al, 2018b). However, the overall worldwide BC emission is estimated to be 4800-7200 Gg per year (Klimont et al., 2017). In 2001, India's total BC emissions were projected to be 1343.78 Gg (Sloss, 2012). Residential fuel burning and transportation contribute maximum to the global anthropogenic BC emission (Helin et al., 2021). About 60 to 80% of residential fuels (coal and biomass) emissions are reported from Asian and African countries, whereas approximately 70% of diesel engine emissions are found

to be from Europe, North America, and Latin America (Johnson et al., 2019; Ayompe et al., 2021; Adeeyo et al., 2022; Sun et al., 2022).

On the other hand, emissions on the Indian subcontinent have increased by 40% since the year 2000 (Kurokawa and Ohara, 2020; Sun et al., 2022). According to Reddy and Venkataraman (2002a, 2002b), the estimated BC emissions in India are fossil fuels, 100 Gg biofuel, 207 Gg open burning, and 39 Gg with a climatic forcing of +1.1 W/m$^2$, black carbon is the second-most significant human emission in the current atmosphere (Sharma et al., 2022). BC concentration was measured by Zhao et al. (2017) in the south-eastern Tibetan Plateau (TP). Daily mean BC loadings ranged from 57.7 to 5368.9 ng/m$^3$ demonstrating a high BC burden even at free tropospheric altitudes (Zhao et al., 2017). Black carbon (BC) deposition was estimated at the Nepal Climate Observatory - Pyramid (NCO-P) site in the Himalayan region during the pre-monsoon season (March-May). A total BC deposition rate of 2.89 µg/m$^3$/day was estimated, resulting in a total deposition of 266 µg/m$^3$ for March–May (Yasunari et al., 2010). From the Indian perspective, several key short-term incidents contribute to a rise in India's BC concentration from biomass burning and other sources (Kumar et al., 2020a). Burning agricultural waste (stubble) is widespread in India and several other nations. Many studies suggest that increased BC in northern India, notably the Indo-Gangetic Plain (IGP) is the global absorbing aerosol hotspot (Venkataraman et al., 2006; Ramanathan and Carmichael, 2008). In India, post-monsoon paddy crop waste burning occurs in the months of October and November in the north and northwest parts of India (Venkataraman et al., 2006). In the north-western Indo-Gangetic Plain (IGP) (especially- Punjab, Haryana, and western Uttar Pradesh), stubble burning is a popular practice (Venkataraman et al., 2006). Long-distance transport of BC aerosols, mostly from Asia to the North Pacific and South America to the southwest Atlantic, is often recognized as a significant factor in local concentration (Evangelista et al., 2007). However, in India, only local sources (89%) affect BC concentrations (Zhang et al., 2013), as there aren't many movements of transboundary aerosols contribution over the IGP (Kumar et al., 2018a; Kedia et al., 2014; Ramachandran and Rupakheti, 2022; Ramachandran et al., 2020). Both marine and continental air masses contributed to total aerosol loading over middle-IGP (Kumar et al., 2017; Shukla et al., 2022).

Black carbon is a light-absorbing particle that is released into the atmosphere directly in the form of ultrafine (<0.1 µm) to fine particles (<2.5µm) (Gupta et al., 2017). BC is a good tracer for particle deposition as it is non-volatile, insoluble, and chemically inert, and it can also mix well with other aerosol species in the atmosphere (Kiran et al., 2018). As a result, BC deposition data are important not just for BC sinks but also for a broader understanding of

aerosol deposition. BC emissions are mostly influenced by significant changes in the energy sector, fuel usage, industrial expansion, and an increase in the number of vehicles (Bisht et al., 2015). Residential fuels like wood, agricultural waste, and cow dung used for cooking and biomass usage for home purposes are the primary sources of BC emissions (Venkataraman et al., 2006). The Asian mainland is a substantial contributor to global BC emissions and has been identified as a hotspot (Gupta et al., 2017). BC has a high absorption ability, accounting for 90-95 percent of total atmospheric aerosol absorption (Hansen et al., 1984). It can absorb solar energy in the visible-infrared band and warm the environment. In comparison to carbon dioxide, BC has a much shorter life cycle in the atmosphere. As a result, mitigation or reduction has a greater positive impact on the atmosphere (Kirchstetter et al., 2004; Takemura and Suzuki, 2019). Changing land use land cover (LULC) has a very significant impact on weather, climate, and aerosols (Mahmood et al., 2010). It is well-established fact that the LULC change has a direct relation with land surface temperature, vehicular emission, and anthropogenic activity (Aithal and MC, 2019). This motivated the present study for further analysis of Sikkim region land use land cover change and its relation with temperature and BC/BrC for March 2021 to March 2022. The current study's objectives are to assess the intra-annual variability of Black/Brown Carbon (BC/BrC) (diurnal/daily/monthly) during the study period March-2021 to March-2022, as well as the interrelationship between meteorological conditions and BC/BrC, along with LULC change for three decades 2000, 2010, and 2020, and its relationship with anthropogenic activity over Gangtok.

**2.0 Study location**

The Gangtok Municipal Corporation (GMC) has been selected for the present study on the basis of its urban exposure and settlement change for three decades as well as congruently temperature rise (Figure S1). The sampling was carried out at the Pani House area in Gangtok, GMC, having a longitude of 88.609°E and a latitude of 27.323°N. Sikkim is surrounded by Nepal, China, and Bhutan from west, north, and east respectively, and consists of the trans and greater Himalayan range. Moreover, Sikkim has one of the most fragile forest covers. However, Gangtok is a densely populated city and capital of the state of Sikkim which is situated in the East Sikkim district (see Figure 1a). The population of Sikkim has been found to have increased as per the Indian census for three decades as can be seen in table S1.

**3.0 Data and Methodology**

The real-time sampling of BC was carried out from 10[th] March 2021 to 17[th] March 2022, at Gangtok using the seven-channel dual spot Aethalometer (Model AE-33-7, Magee Scientific,

USA). The Aethalometer AE-33 is an aerosol instrument with a detection limit of <0.005 µg/m³ for a 1-hour period and a measuring range of 0.01 to 100 µg/m³. It has a programmable measuring frequency of 1 second or 1 minute and a programmable flow rate of 2 to 5 lpm. The data was collected for the measurement of BC and BrC associated with particulate matter having an aerodynamic diameter of less than 2.5 µm ($PM_{2.5}$). The concentration of BC, BrC, $BC_{bb}$, and $BC_{ff}$ have been estimated by the Carbonaceous Aerosol Analysis Tools (CAAT) software tool from the Magee Scientific Aethalometer model AE33 (Hansen and Schnell, 2005). The carbon dioxide ($CO_2$) was measured using a $CO_2$ sensor (Vaisala-GMP343) which is attached to the aethalometer. The inlet of the aethalometer was mounted at a height of 15 m above ground level. One of the main sources of uncertainty in utilizing aerosol absorption measurements to estimate the BrC absorption coefficient at 370 nm is the potential contribution of other species, such as black carbon and dust, to the measured absorption. This can result in an overestimation of BrC mass concentration, especially in environments where these species coexist. However, the Sikkim region stands out for having one of the highest precipitation levels globally and minimal dust pollution contribution. Consequently, there is likely to be less over or underestimation. Therefore, in this study, mass concentration was employed to address these uncertainties.

A new data set of BC, BrC, Black Carbon from biomass burning ($BC_{bb}$), Black Carbon from fossil fuels ($BC_{ff}$), the percentage contribution of biomass burning to BC (BB%) and $CO_2$ has been generated over the unreported region of Sikkim Himalaya. The diurnal and monthly data sets of BC, $BC_{bb}$, $BC_{ff}$, BrC, BB%, and $CO_2$ have been given in the details in supplementary materials (Table S2 and S3). In addition to this, the meteorological data has been selected for ERA5 reanalysis for the study. LULC data has been taken from USGS earth explorers of 2000 and 2010 Landsat-5, 2020 Landsat-8, and 2021 for Sentinel-2 (Karra et al., 2021). LULC data has been chosen for the month of December to minimize the cloud cover. The details of the LULC calculation steps used are given in the supplementary section (methodology S1.3). The brief of the data set is discussed in the table 1.

**3.1 Estimation of BrC**

The Carbonaceous Aerosol Analysis Tools (CAAT) software tool from the Magee Scientific Aethalometer model AE33 was utilized to estimate the concentrations of BC, BrC, $BC_{bb}$, and $BC_{ff}$. The absorption coefficients of BC and BrC were determined using the multi-wavelength absorption coefficients provided by the aethalometer. The presence of BrC was identified by observing the maximum light absorption between 370–590 nm, but its absorption may increase

significantly below this range depending on its composition. The attenuation of illumination measured in this study using the aethalometer was attributed solely to the contribution of BC and BrC. It is believed that the absorption coefficient at 370 nm measured by the aethalometer represents the combined absorption coefficients of BC and BrC, which is denoted as $\sigma_{BC+BrC}$ (370 nm). This assumption is similar to the model used in the multi-wavelength absorbance analyzer (MWAA) approach for source allocation, as described in Massabò et al. (2015). Equation (1) was used to calculate the $\sigma_{BrC}$ (370 nm) absorption coefficient (supplementary methodology S1), which involved subtracting the contribution of BC ($\sigma_{BC}$ (370 nm)) from the observed absorption coefficient ($\sigma_{BC+BrC}$ (370 nm)).

$$\sigma_{\mathrm{BrC}}(370\,\mathrm{nm}) = \sigma_{BC+BrC}(370\,\mathrm{nm}) - \sigma_{BC}(370\,\mathrm{nm}) \qquad \text{Eq. (1)}$$

The $\sigma_{BC}$ (370 nm), was calculated by applying the power-law fit to absorption data in the 590-950 nm wavelength range provided in equation (1).

$$\sigma_{\mathrm{BC}}(\lambda) = \beta\,\lambda^{-AAE_{\mathrm{BC}}} \qquad \text{Eq. (2)}$$

The absorption angstrom exponent of BC is denoted as $AAE_{BC}$, with $\beta$ being a constant value. As BC is a significant contributor to light absorption at wavelengths beyond 590 nm, the contribution of other aerosol species can be neglected, and the $AAE_{BC}$ can be calculated using equation (3), as stated in Rathod and Sahu (2022). The AAE for both BC and BrC can be expressed as $\sigma$, and in this study, the AAE definition by Moosmüller et al. (2011a) was used instead of the AAE specified for a wavelength pair. This value is determined by equation (3), which calculates the negative log-log slope of the absorption spectrum at wavelength $\lambda$.

$$\mathrm{AAE}_{BC} = -\frac{d\ln\sigma_{BC}}{d\,\ln\lambda} \qquad \text{Eq. (3)}$$

Instead of the conventional approach where $AAE_{BC}$ is assumed to be 1, we utilized the $AAE_{BC}$ that was observed onsite to calculate $\sigma_{BC}(\lambda)$. Equation (4) was employed to determine $\sigma_{BrC}$ (370 nm) by substituting $\sigma_{BC}(\lambda)$ at 370 nm, which was obtained using equation (2) (Wang et al., 2020), into equation (4) (refer to supplementary methodology S1.1, S1.2, and Figure S2 for details).

$$\sigma_{BrC}(370\,\mathrm{nm}) = \sigma_{\mathrm{BC+BrC}}(370\,\mathrm{nm}) - \beta(370nm)^{-AAE_{BC}} \qquad \text{Eq. (4)}$$

To calculate $\sigma_{BrC}(\lambda)$ at 470 nm and 520 nm, we can subtract the modelled BC from the measured absorption coefficients, in a similar manner. It is worth noting that the BrC absorption coefficients are very low at wavelengths beyond 590 nm (Wang et al., 2020),

according to Rathod et al. (2017) and Rathod and Sahu (2022), hence they are not taken into
account (supplementary methodology S1; referred to equations S1 to S13).
**3.2 Data Analysis**
LULC change also has a direct impact on vehicular emissions and other anthropogenic
activities. Urbanization, conceivably, can lead to increased vehicle traffic and emissions,
which can contribute to air pollution and climate change. Changes in land use can also affect
the amount and type of vegetation, which can influence the carbon cycle and the amount of
greenhouse gases in the atmosphere. The ERA-5 reanalysis data has been used for
meteorological analysis viz. wind pattern, precipitation, relative humidity, and temperature
(Hersbach et al., 2020). The hourly data has been taken for the analysis and then the daily,
monthly, and seasonal average has been computed for the study period over the Sikkim and
surrounding states for a better understanding of the meteorological conditions influencing the
BC, and BrC. The total precipitation is computed as a sum of the hourly data for a day to daily
total precipitation and further, it was summed for monthly cumulative total precipitation using
the sum formula as
$$\text{Monthly Cumulative Total Precipitation} = \sum_i^n X \qquad \text{Eq. (5)}$$
Where 'i' is the initial 'n' the last date and X is the hourly total precipitation taken from ERA5.
The wind circulation has been computed using the u-component and v-component of wind and
the wind speed has been calculated as
$$\text{Wind Speed} = \sqrt{u^2 + v^2} \qquad \text{Eq. (6)}$$
The temperature and relative humidity averaged have been computed using the mean formula
as
$$Average = \frac{\sum_i^n X}{n} \qquad \text{Eq. (7)}$$
Where, 'i' is the initial and 'n' last date of the variables such as temperature, relative
humidity, and wind components.
Let x and y be two real-valued random variables such that the correlation coefficient Spearmen
Pearson can be calculated between the BC/BrC and meteorological parameters. The
Coefficient of Pearson Correlation (PCC) (Pearson, 1909; Benesty et al., 2009) as
$$PCC = \frac{n(\sum xy) - (\sum x)(\sum y)}{\sqrt{[n \sum x^2 - (\sum x)^2][n \sum y^2 - (\sum y)^2]}}$$
Eq. (8)

Where 'n' is the population size of the variables used for the study.
Table 1 contains additional information about the dataset, and a more detailed methodology
can be found in the supplementary section (S1).

## 4.0 Results and Discussions

The anthropogenic activities in Gangtok have drastically increased in the last 20 years. As
evident from Figures 1b, c, and d, LULC has been changed from 2000 to 2020 over the
Gangtok Municipal Corporation (GMC). Population change and growth have also been
observed in the Sikkim (Table S1). LULC during the years 2000 and 2010 evidently shows
that most of the fallow land has been built up due to a recent change in the policy of
construction in Sikkim suggesting urban settlement load over Gangtok has increased
significantly. As a result, there is a significant increase in built-up areas in GMC for the last
20 years. The vegetation cover has also reduced from 2000 to 2020 (Figure 1b, c, and d). The
rainfed water bodies are reducing from the GMC. However, due to its seasonal nature, streams
are lesser emerged in 2020 which perhaps shows the precipitation pattern alteration over GMC
due to the highly built-up sprawl.  The built-up extent has been sprawling and consuming the
dense vegetation regions as well. This increases the study region's urge to be acknowledged so
that Sikkim's future policymakers can consider the effects of rising anthropogenic activities.
This anthropogenic activity leads to a heavy load on the environment over one of the cleanest
states of India. Long-term spatiotemporal variation of 2-meter air temperature justifies the
LULC change and warming pattern (Xiao-lei et al., 2022) over the Gangtok region (Figure
S1a, S1b, S1c, S1d, and S1e). The decadal warming rate is varying from $0.25^O$ to $0.45^OC$
(Figure S1e). Thereafter, BC and BrC over the Gangtok have been measured to report the issue
and get more attention to the scientific and local community. The higher anthropogenic activity
releases a higher amount of emission in the name of development due to the population load
on the region (Shaddick et al., 2020) (i.e., the growth rate has been raised from 12.89 to 13.05%
in recent years) (Table S1). Diurnal variation of the BC, BrC, $BC_{bb}$, $BC_{ff,}$ and $CO_2$ show two
peaks. BC, $BC_{ff,}$ and $CO_2$ have almost similar time of peaks observed. The first peak is found
during 8-10 AM. And, the second peak is observed during 8-10 PM. However, BrC and $BC_{bb}$
have the peak concentration during 10-11 AM and 6-8 PM (Figure 2a), suggesting the peak
biomass burning time over the region. The meteorological conditions are observed as low
dewpoint, low temperature, high surface pressure, low wind speed, and high relative humidity
to the corresponding 8-10 AM, while the opposite is found in 8-10 PM referred to Figure 2b.
The daily time series of the BC, $BC_{bb}$, $BC_{ff}$, BrC, BB%, and $CO_2$ show the highest fluctuation
from $20^{th}$ to $30^{th}$ March in both 2021 and 2022 years respectively. The maximum BC (BrC)
content was found in March 2022 (April-2021), at 43.5µg/m³ (32µg/m³). The lowest
fluctuation is observed from $15^{th}$ May to $15^{th}$ September 2021 (Figure 3a). The intense peaks
of BC, $BC_{ff,}$ and $CO_2$ were observed from $10^{th}$ October to $15^{th}$ November 2021 (Figure 3a)
which may be linked to the heavy tourist season of the state and indicate the traffic overload
in the Gangtok (Sharma et al, 2022). The meteorological conditions also favour similar
circumstances to accumulate the pollutant from $10^{th}$ October to $15^{th}$ November 2021 (Figure
3b). The lowest surface pressure with minimum fluctuation and the highest temperature and
dewpoint temperature with minimum fluctuation was noticed from the $15^{th}$ June to $20^{th}$
September 2021 (Figure 3b). BrC is found to be the highest with significant variability from
the $10^{th}$ of January to the $30^{th}$ of March, pointing to winter wood burning for livelihood, which
is also supported by $BC_{bb}$. The monthly variations of BC, $BC_{bb}$, $BC_{ff}$, BrC, and BB% are
discussed in Figure 4a, and the highest value of standard deviation was observed during March
2022 for BC, $BC_{ff}$, and April 2021 for $BC_{bb}$, BrC, and BB%. The $CO_2$ is observed almost
constant with a small value of standard deviation. The maximum concentration of the BC, $BC_{ff}$
is found in March 2022. However, $BC_{bb}$ and BrC were measured highest in April 2021. This
is probably inferring to high tourist season (i.e., vehicular emission) as well as random wood
burning at higher altitude regions surrounding the Gangtok. The minimum concentration of
the BrC was seen in the month of August 2021 as the highest total precipitation month with
high wind speed, temperature dewpoint temperature, and relative humidity (Figure 4b, S3, and
S4) (Rana et al., 2023).
The good correlation between BC and $BC_{ff}$ showed that the primary source of BC is fossil fuel
combustion (Osborne et al, 2008; Jung et al., 2021). A significant correlation between $BC_{bb}$
and BrC indicates that biomass burning is a major contributor to BrC (Prabhu et al., 2020),
which is supported by the BB% and BrC (Figure 5). The positive correlation between $CO_2$ and
BC/$BC_{ff}$ suggests that fossil fuel burning is influencing the $CO_2$ concentration (Rana et al.,
2023). Dewpoint temperature and $CO_2$ have a significant positive correlation suggesting
positive radiative forcing of $CO_2$ (Huang et al., 2017; Stjern et al., 2023). A similar relationship
has also been observed for temperature. $BC_{bb}$/BrC and temperature have a significant negative
correlation suggesting the negative radiative nature of the $BC_{bb}$/BrC (Figure S5). Moreover,
net thermal/solar radiation (STR/SSR) and BC/BrC have a significant positive correlation
(Figure 5, and S5) (Liu et al., 2020). A significant positive correlation between surface pressure
and BC/$BC_{ff}$ ($BC_{bb}$/BrC) has been observed (Figure 5). Higher surface pressure creates calm
conditions and a stable boundary layer, which keeps the pollutants accumulated in the
boundary layer (Igarashi et al., 1988; Lee et al., 1995; Bharali et al., 2019; Liu et al., 2021).
However, the opposite has been observed for the wind indicating the dispersion of pollutants
with a strong negative correlation. A similar relationship has been observed between total
precipitation and all the pollutants, indicating the process of wet scavenging of pollutants (Yoo
et al., 2014; Ohata et al., 2016; Ge et al., 2021; Wu et al., 2022). The relative humidity is also
showing a similar result to the total precipitation with greater values of coefficient. The
negative correlation between total precipitation and surface pressure suggests that the rain falls
over the region mostly occurs in a low-pressure system that is caused due to the vertical rising
of an air parcel and causes condensation and precipitation (Johnson and Hamilton, 1988;
Sarkar, 2018). Aerosols, including black carbon (BC) and absorbing organic aerosol (brown
carbon, BrC), play a vital role as condensation nuclei for cloud-droplet growth, and a fraction
of mineral particles initiate the freezing of supercooled cloud droplets, leading to the release
of precipitation in the form of snow, hail, and rain (Mason, 1999). However, cloud
condensation nuclei formation and precipitation are prompted by primary aerosols, secondary
aerosols (such as nitrate, and sulfate), and BC/BrC (Ohata et al., 2016; Liu et al., 2020; Moteki,
2023). Moreover, BC particles are mainly hydrophobic and less efficient as CCN compared to
more hydrophilic particles; they can still act as CCN under certain conditions. These conditions
include the size and mixing state of the particles, as well as the atmospheric conditions such
as relative humidity and temperature (Ohata et al., 2016; Moteki, 2023; Liu et al., 2020). The
conditions required for BC particles to efficiently play the role of CCN depend on several
factors, including their size, mixing state, and atmospheric conditions (Moteki, 2023; Liu et
al., 2020). For example, smaller BC particles are more efficient as CCN than larger ones
(Moteki, 2023). The mixing state of BC particles also plays a role, as externally mixed BC
particles are less efficient as CCN than internally mixed ones (Liu et al., 2020). Atmospheric
conditions such as relative humidity and temperature also affect the efficiency of BC particles
as CCN (Moteki, 2023). For example, higher relative humidity and lower temperatures can
increase the efficiency of BC particles as CCN (Moteki, 2023). Additionally, relative humidity
over the study region is very high during the entire year with the favourable temperature.
Thereafter, BC and BrC have a crucial role in the precipitation mechanism (Zhu et al., 2021;
Li et al., 2023a) over the study region. Total precipitation and wind circulation indicated that
the study region received precipitation throughout each month of the study period (i.e., most

of the time in the form of rain and occasionally snow). Hence the maximum is observed in August and the minimum in March 2022. The wind pattern illustrates the monsoon seasonal strong influence from May to September 2021 (Figure 6). The wind converses in the valley and diverges from the mountain for the rest of the period (figure 6). Because the strong wind and heavy rainfall indicated pollution scavenging (rain out or wash out), it is significantly negatively correlated as TP vs $BC_{bb}$; TP vs $BC_{ff}$; TP vs BrC (Figure 5).

The relative humidity and temperature follow the same pattern when the temperature gradients change from January to December, resulting in a decrease in moisture content in the atmosphere (Figure S6). The lowest in the month of February is observed and the temperature gradient gets steep from November (Figure S6). The dewpoint temperature contour and surface pressure shading match well suggesting that the surface pressure creates the dewpoint temperature gradient and keeps it sustained and stable atmospheric condition (Jung et al., 2023) (Figure S7). During the month of June, it is very peculiar that the dewpoint temperature contours are wide and a very small gradient is observed (Figure 7). This points toward the warm conditions during the June over entire Sikkim. The cloud cover and convective precipitation over Sikkim are discussed in Figure 7. It is clear from (Figures 7a to d) that the region is not receiving much convective precipitation even if there is huge cloud cover, which leads to a conclusion of orographic precipitation over the region (Figure 7). However, the relative humidity is very high over the sampling site from the lower to upper middle level of the atmosphere during the study period (Figure S3). Most of Sikkim receives convective rain from May to September, which indicates that the region has strong convective activity added from the Bay of Bengal during the monsoon season (Rahman et al., 2012; Kumar et al., 2020b; Kakkar et al., 2022; Biswas and Bhattacharya, 2023). Again, from October to April, the region does not receive convective rain even though there is strong cloud cover pointing toward the orographic rainfall over the entire Sikkim (Kumar and Sharma, 2023). That's making the Sikkim unique weather conditions (Figures S3 and S4). And, the least concentration of BC, $BC_{ff}$, $BC_{bb}$, and BrC is observed during the monsoon months. This observation supports the convective rain, as rain out scavenging, of all pollutants (Liu et al., 2020; Moteki, 2023). During the monsoon season, the region experiences high convective activity, which is added from the Bay of Bengal (Brooks et al., 2019; Liu et al., 2020; Moteki, 2023; Sankar et al., 2023). Convective rain is an effective process for removing air pollutants from the atmosphere (Liu et al., 2020; Moteki, 2023). Wet removal of BC and BrC occurs via cloud particle formation and subsequent conversion to precipitation or impaction processes with hydrometeors below clouds during precipitation (Liu et al., 2020; Moteki, 2023; Sankar et al.,

2023). The BC and BrC have a significant positive correlation with thermal and solar radiation (Zhang et al., 2020; Wang et al., 2021; Li et al., 2023a). A stronger negative correlation between $CO_2$ and surface thermal radiation (STR) and surface solar radiation (SSR) would have significant implications (Figure 5). The negative correlation between $CO_2$ and STR implies that as the concentration of $CO_2$ in the atmosphere increases, the amount of heat radiating from the Earth's surface into space decreases (Zhang et al., 2020). This can lead to an increase in the Gangtok's temperature, which can have various impacts on climate and weather as well (Figures S1, and 5). The negative correlation between $CO_2$ and SSR implies that as the concentration of $CO_2$ in the atmosphere increases, the amount of solar radiation absorbed by the Earth's surface decreases (Davis, 2017; Zhang et al., 2020; Li et al., 2023b) (Figure 5). Overall, a significant negative correlation between $CO_2$ and STR/SSR would indicate a stronger influence of greenhouse gas concentrations on the surface's radiation balance (Chiodo et al., 2018) and would have important implications for climate change as well as anomalous warming over the Gangtok region (Figure S1).

**5.0 Conclusions**

In accordance with the LULC between 2000 and 2010, Sikkim's recent changes to its development regulations have resulted in the majority of fallow land being consumed by construction, which suggests that Gangtok's urban settlement load has increased significantly. In addition, the LULC for 2020 depicts a booming built-up region over the GMC. From 2000 to 2020, the vegetation cover has likewise decreased. However, due to the seasonal nature, streams are lesser in 2020, indicating precipitation pattern variation over GMC. The areas covered in dense vegetation are also being consumed by the expanding built-up area. The present study is the report of newly produced data BC and BrC for the fragile region of the Himalayas and its relation with meteorological conditions. It has been observed that the temperature over Gangtok is increasing as well. The peak concentration of BC/BrC has been found during October 2021, March 2021, and 2022. The diurnal distribution of BC/BrC suggests the two peaks in a day, first at 8-10 AM and second at 9-11 PM. The meteorological conditions for the same have been observed to be favourable to diurnal variation of BC/BrC concentration. The monthly variation of the BC/BrC delineated the peak concentration of BC, $BC_{bb}$, and $BC_{ff}$, during March 2022. However, BrC and BB% have maximum concentration during April 2021. BB% and BrC as well as BB and carbon dioxide have a strong significant positive correlation coefficient, which is evidence that biomass burning is a substantial factor in the rise in carbon dioxide levels. In addition to this, there is a strong, positive correlation between $CO_2$ and BC/$BC_{ff}$, indicating that burning fossil fuels is also one of the causes of

rising $CO_2$ levels. The net thermal radiation, net solar radiation, and BC, BrC relationship
suggested that BC and BrC have positive radiative forcing. Furthermore, the monsoon months
show the lowest concentrations of BC, $BC_{bb}$, $BC_{ff}$, BrC, and BB%, demonstrating the
convective rain (i.e., rain out scavenging) ability to remove a majority of contaminants. BC
particles in the atmosphere have a strong ability to absorb solar radiation, and their lifetime
depends on atmospheric transport, aging, and wet scavenging processes. Organic aerosols,
including BrC, can undergo photochemical aging, affecting their ability to act as cloud
condensation nuclei (CCN). The effective density of BC is a crucial factor in evaluating its
climate effect, and variations in BC density can lead to uncertainties in predicting CCN number
concentration.

**Data Availability**

Data is provided in the 'supplementary section' and for further detail knowledge about it can
be available from the corresponding author on the adequate request.
Data link for the data access:
https://docs.google.com/spreadsheets/d/1N4F_fT68syY6n0UIfA6nzI5o-
8LUWjyFfk5NpfquRyg/edit?usp=sharing

**Conflict of Interest**

None conflict of interest.

**Authors Contribution**

Dr. Pramod Kumar: conceptualization, drafting, writing, figures, and editing
Ms. Khushboo Sharma: sampling, data analysis, and figures.
Ms. Ankita Malu: data analysis, figures, and editing
Mr. Rajeev Rajak: editing
Ms. Aparna Gupta: editing
Mr. Bidyutjyoti Baruah: editing
Mr. Jayant Sharma: sampling
Dr. Shailesh Yadav: editing, and mentoring
Dr. Thupstan Angchuk: editing, and mentoring
Dr. Nishchal Wanjari: editing and mentoring.
Dr. Anil Kumar Misra: editing and mentoring.
Dr. Rakesh Kumar Ranjan: conceptualization, data interpretation, mentoring, and editing.

## Acknowledgments

Authors acknowledge to the Department of Science and Technology, Government of India, and host department "DST's Centre of Excellence (CoE), at Department of Geology, Sikkim University, DST/CCP/CoE/186/2019 (G)," for the generation of BC/BrC data. We also acknowledge to free data sources used in the study as ERA5, and USGS earth explorer. Authors appreciate freely available software such as R-studio, QGIS, CDO, and GrADS used for the analysis and visualization. We also acknowledge Anirud Rai, Kuldeep Dutta, Abhinav Tiwari, Richard Rai, and the anonymous persons who so ever have helped and supported the Black Carbon data collection.

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

**List of Figures**

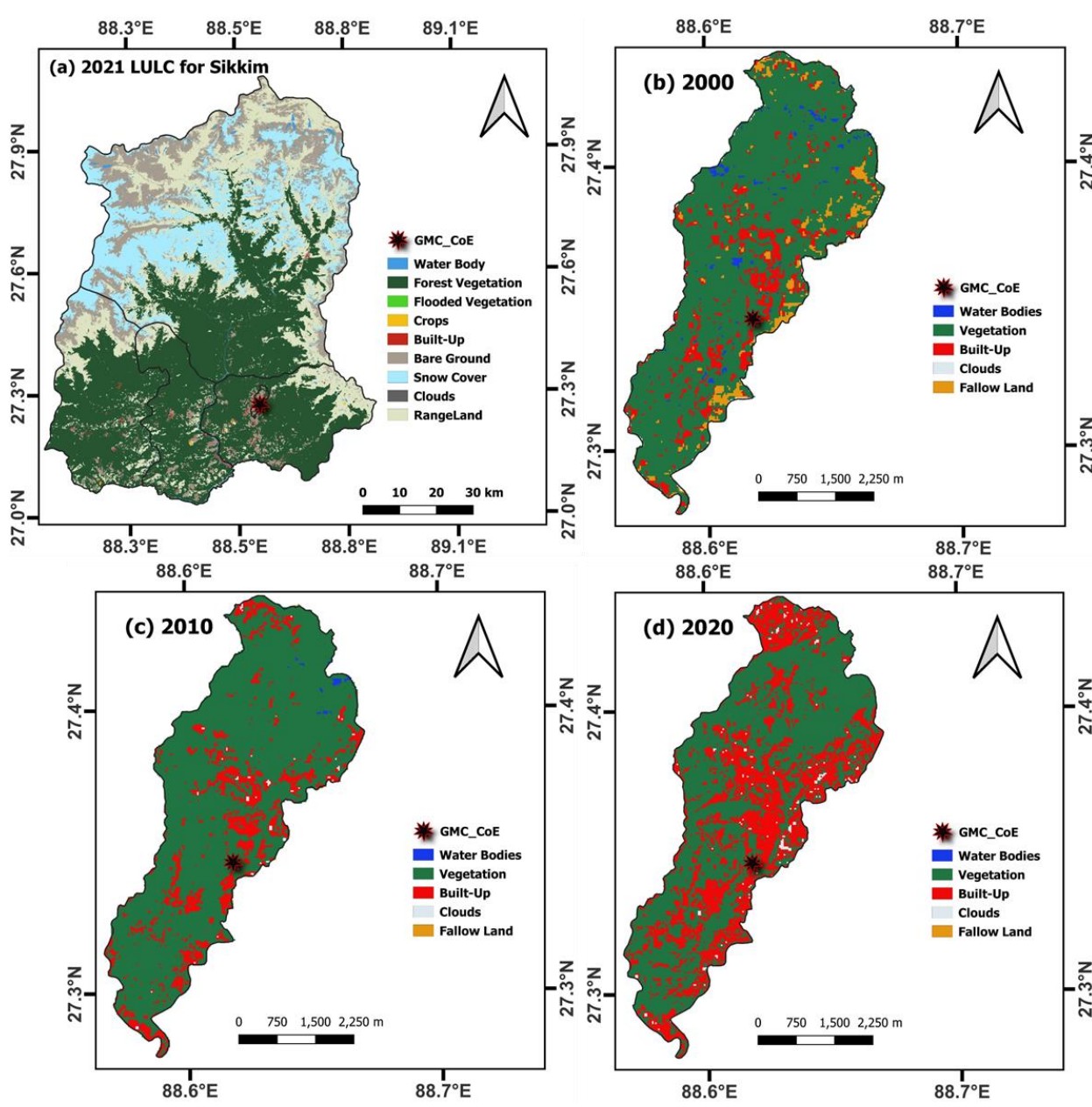


Figure 1. The study location and land use land cover for 2000, 2010, 2020, and 2021 for December over Gangtok and Sikkim region using Landsat-5, Landsat-8, and Sentinel-2 data sets.

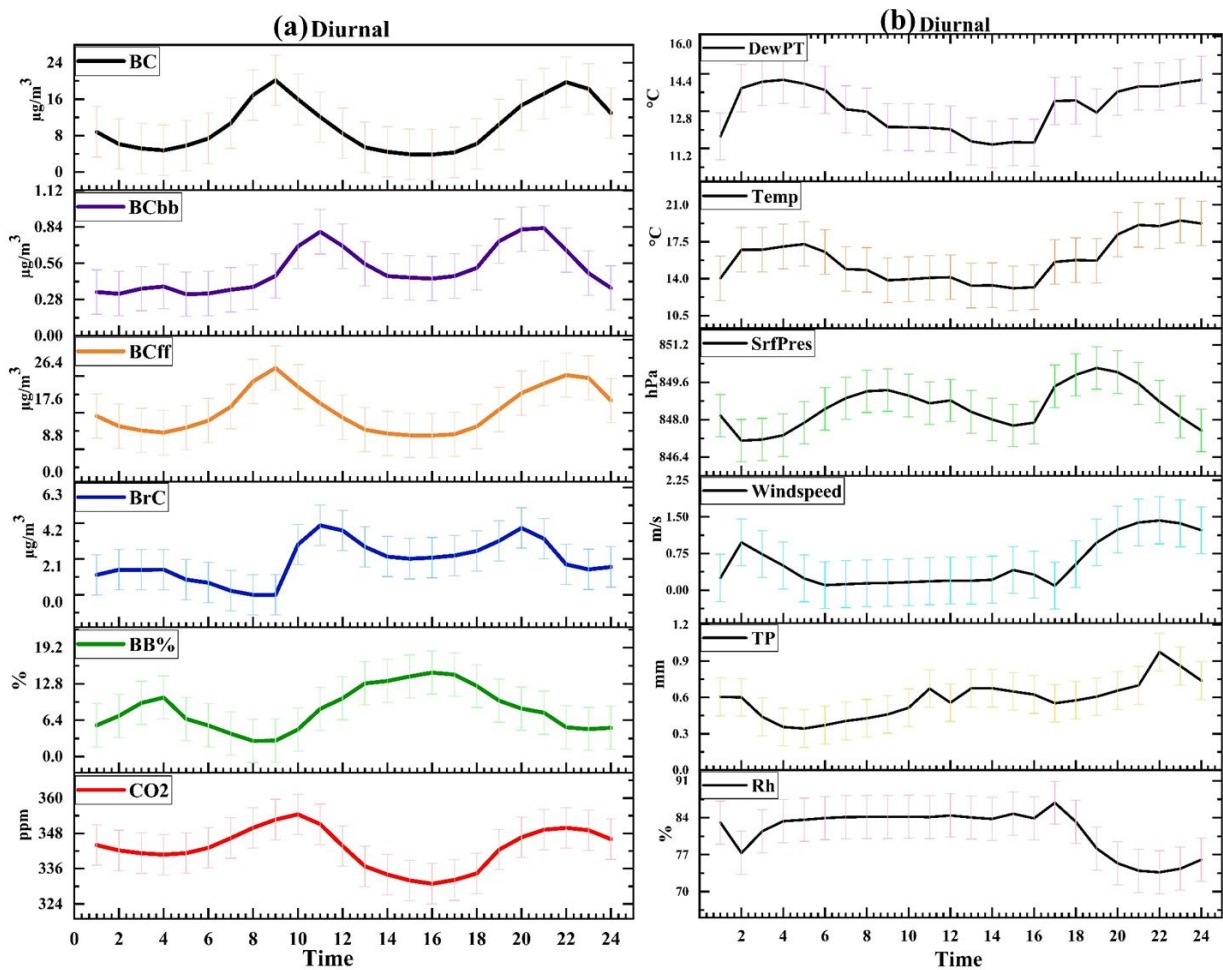


Figure 2. (a) The hourly observation of Black Carbon, Black Carbon through biomass burning,
Black Carbon through fossil fuel, Brown Carbon, Biomass Burning percentage and Carbon
Dioxide (BC, $BC_{bb}$, $BC_{ff}$, BrC, BB%, and $CO_2$, respectively) (The corresponding unit for BC,
$BC_{bb}$, $BC_{ff}$, BrC: µg/m³; BB%: % and $CO_2$: ppm) for 16th March 2021 to 10th March 2022 over
study location (lat:27.32; lon:88.61). The light colour shading refers to ±σ standard deviation
for each variable. (b) Same as Figure 2a, but for meteorological parameters such as dewpoint
temperature (DewPT), temperature (Temp), surface pressure (SrfPres), windspeed, total
precipitation (TP), and relative humidity (Rh) from 16th March 2021 to 10th March 2022.

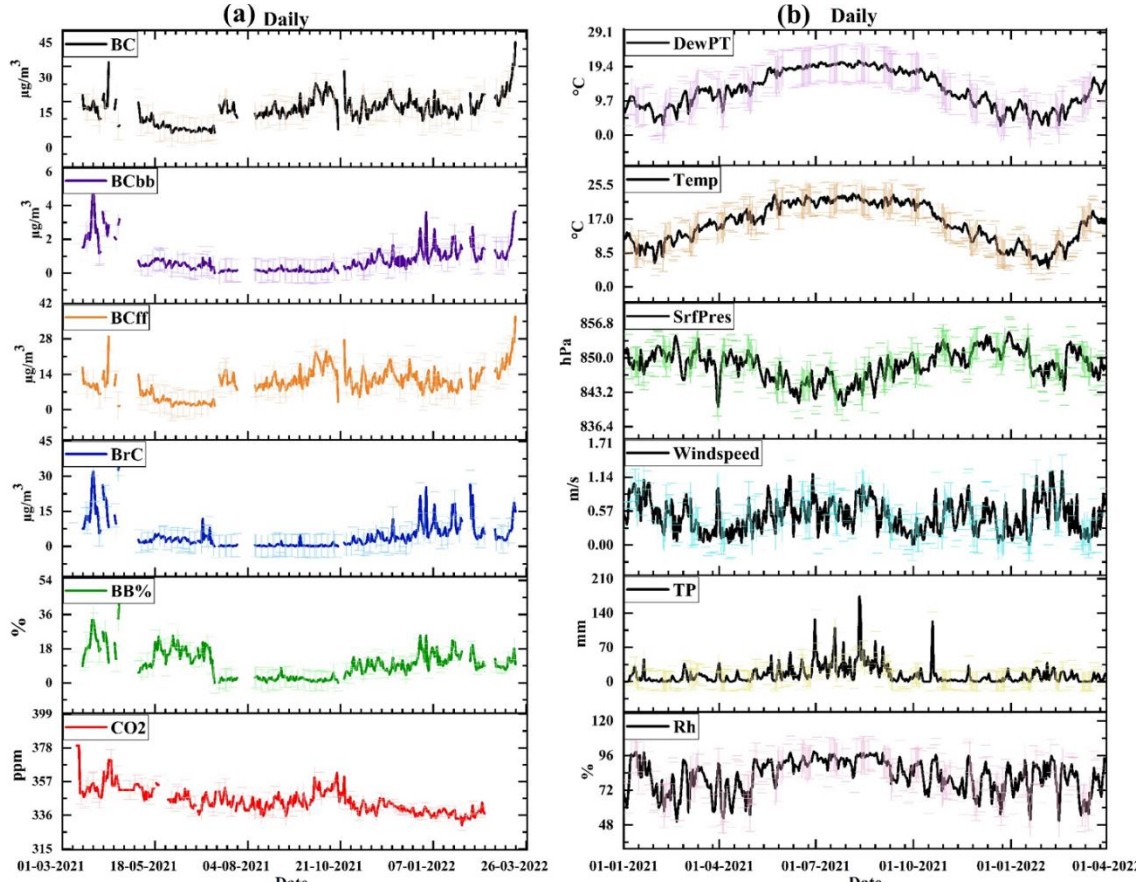


Figure 3. (a) The daily mean of Black Carbon, Black Carbon through biomass burning, Black Carbon through fossil fuel, Brown Carbon, Biomass Burning percentage and Carbon Dioxide (BC, $BC_{bb}$, $BC_{ff}$, BrC, BB%, and $CO_2$, respectively) (The corresponding unit for BC, $BC_{bb}$, $BC_{ff}$, BrC: μg/m3; BB%: % and $CO_2$: ppm) for 16[th] March 2021 to 10[th] March 2022 over study location (lat:27.32; lon:88.61). The light colour shading refers to ±σ standard deviation for each variable. (b) same as Figure 3a, but for meteorological parameters such as dewpoint temperature (DewPT), temperature (Temp), surface pressure (SrfPres), Windspeed, total precipitation (TP), and relative humidity (Rh) from 1[st] January 2021 to 31[st] March 2022.


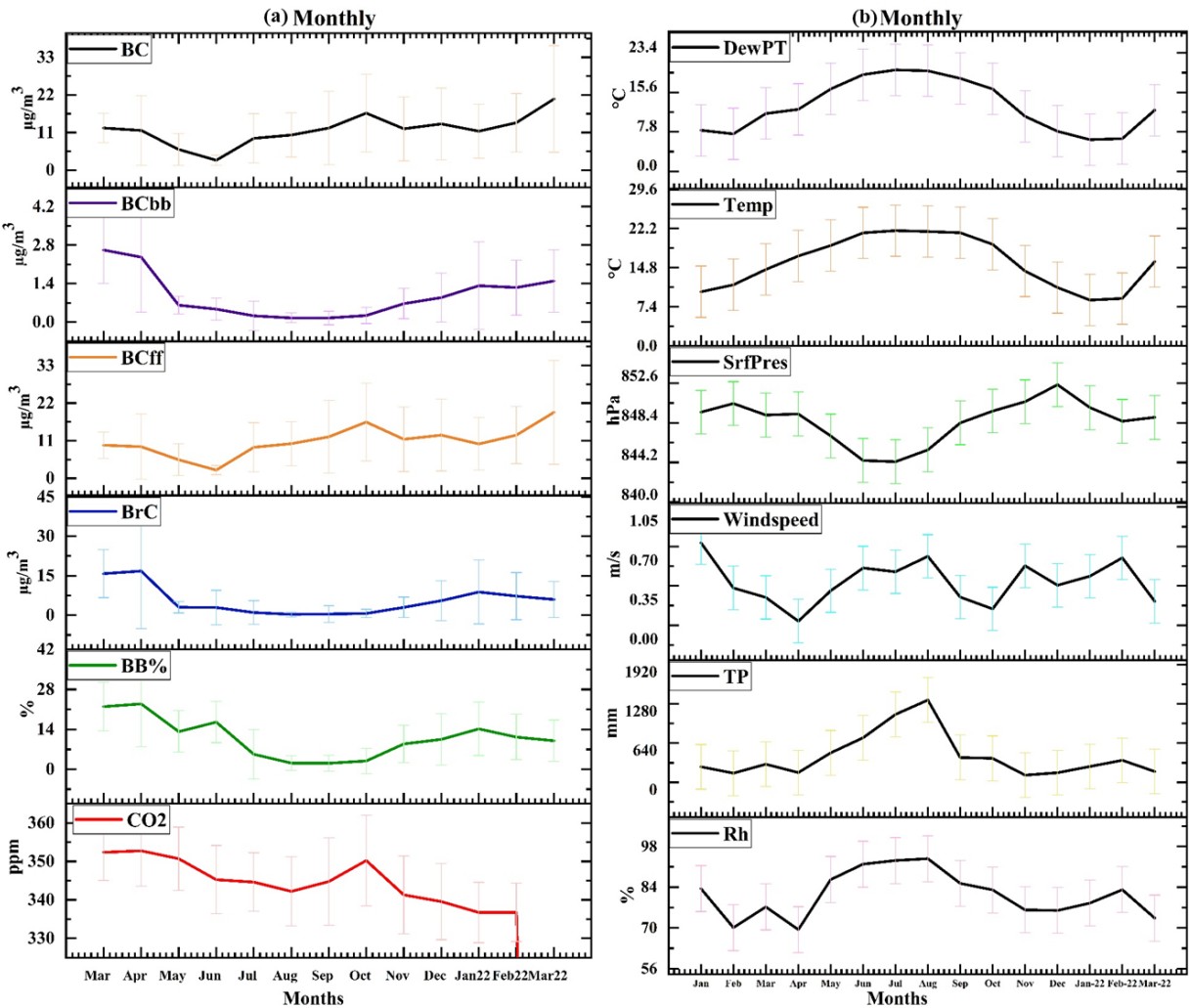


Figure 4. (a) The monthly mean of Black Carbon, Black Carbon through biomass burning,
Black Carbon through fossil fuel, Brown Carbon, Biomass Burning percentage and Carbon
Dioxide (BC, $BC_{bb}$, $BC_{ff}$, BrC, BB%, and $CO_2$, respectively) (The corresponding unit for BC,
$BC_{bb}$, $BC_{ff}$, BrC: µg/m3; BB%: % and $CO_2$: ppm) for 16th March 2021 to 10th March 2022 over
study location (lat:27.32; lon:88.61). The error bar shows ±σ standard deviation for each
variable. (b) Same as Figure 4a, but for meteorological parameters such as dewpoint
temperature (DewPT), temperature (Temp), surface pressure (SrfPres), windspeed, total
precipitation (TP), and relative humidity (Rh) during January 2021 to March 2022.

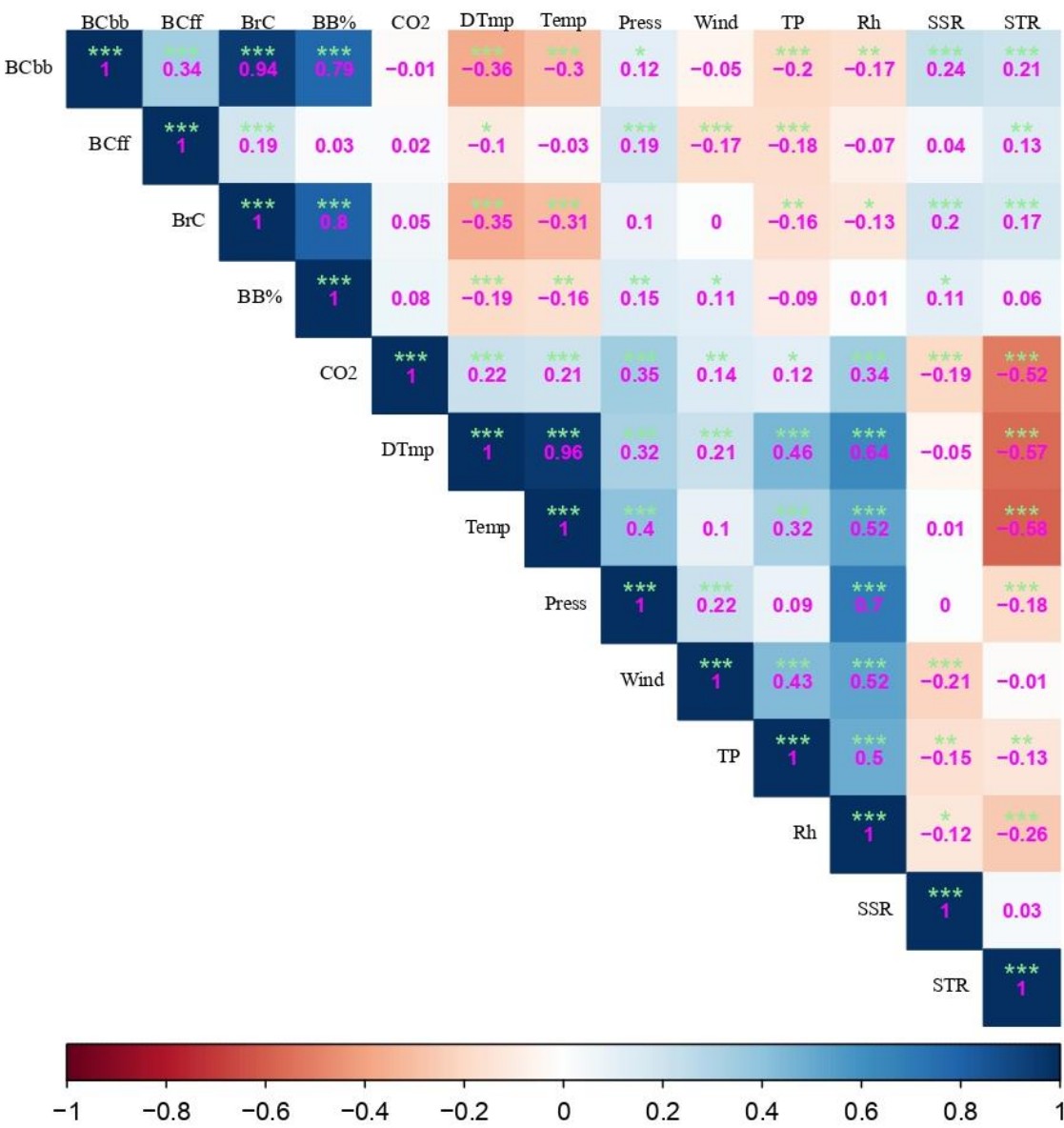


Figure 5. Correlation among BC, $BC_{bb}$, $BC_{ff}$, BrC, BB%, $CO_2$ and, dewpoint temperature (DTmp), temperature (Temp), surface pressure (Press), Wind, total precipitation (TP), Relative humidity (Rh), net solar radiation (SSR), and net thermal radiation (STR). The (***) shows 99% significance, (**) shows 95% significance, (*) 90% significance, and () shows no significance. The correlation coefficient values (-0.3 to -0.49) or (0.3 to 0.49) are considered 'a good correlation', and values $\leq$ (-0.5) or $\geq$ (0.5) are considered "a strong correlation".

779

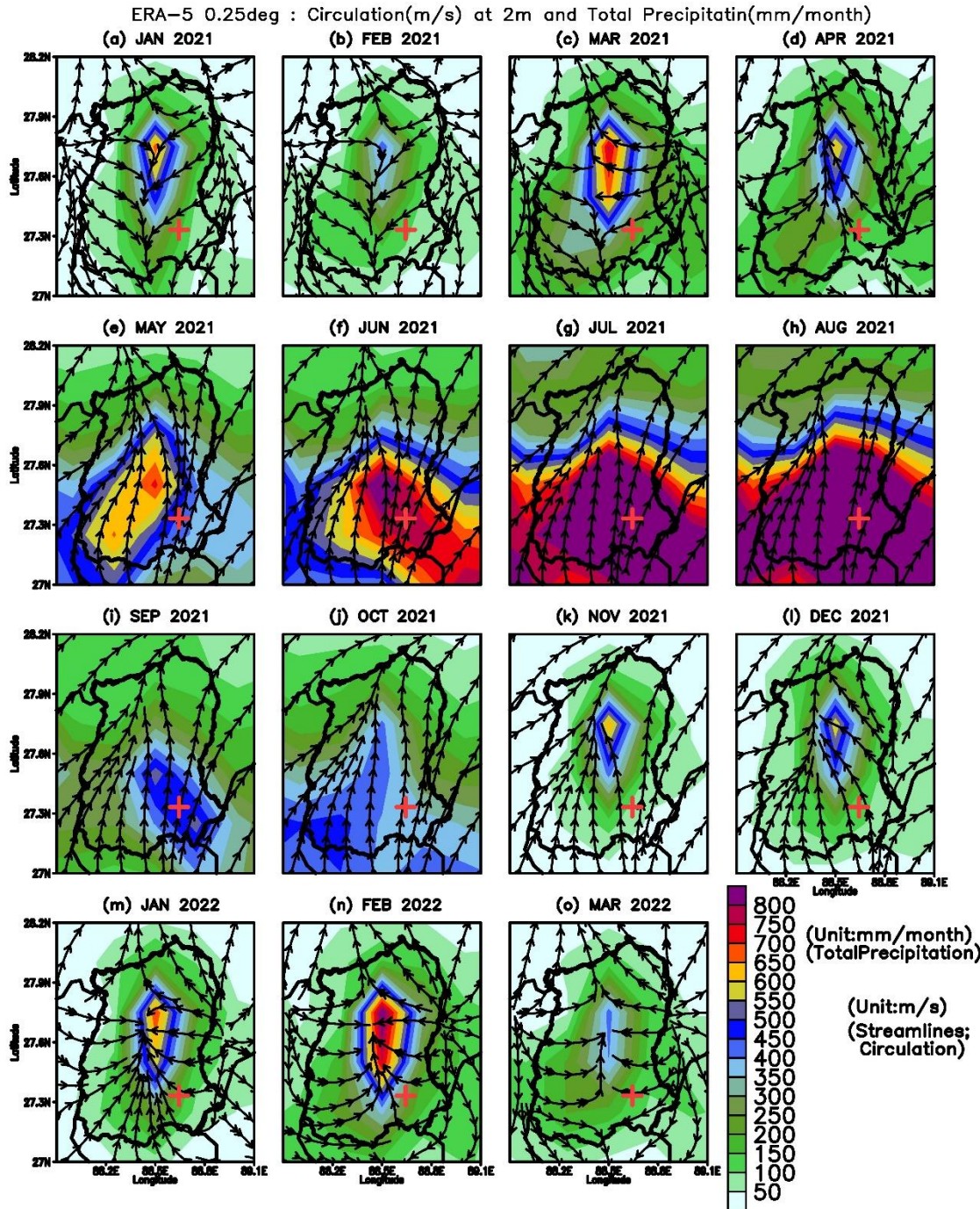

Figure 6. Monthly total precipitation (cumulative) and wind circulation pattern during January 2021 to March 2022. The shading shows precipitation patterns, and the streamline shows wind circulation. The (+) mark is a representation of the sampling location.

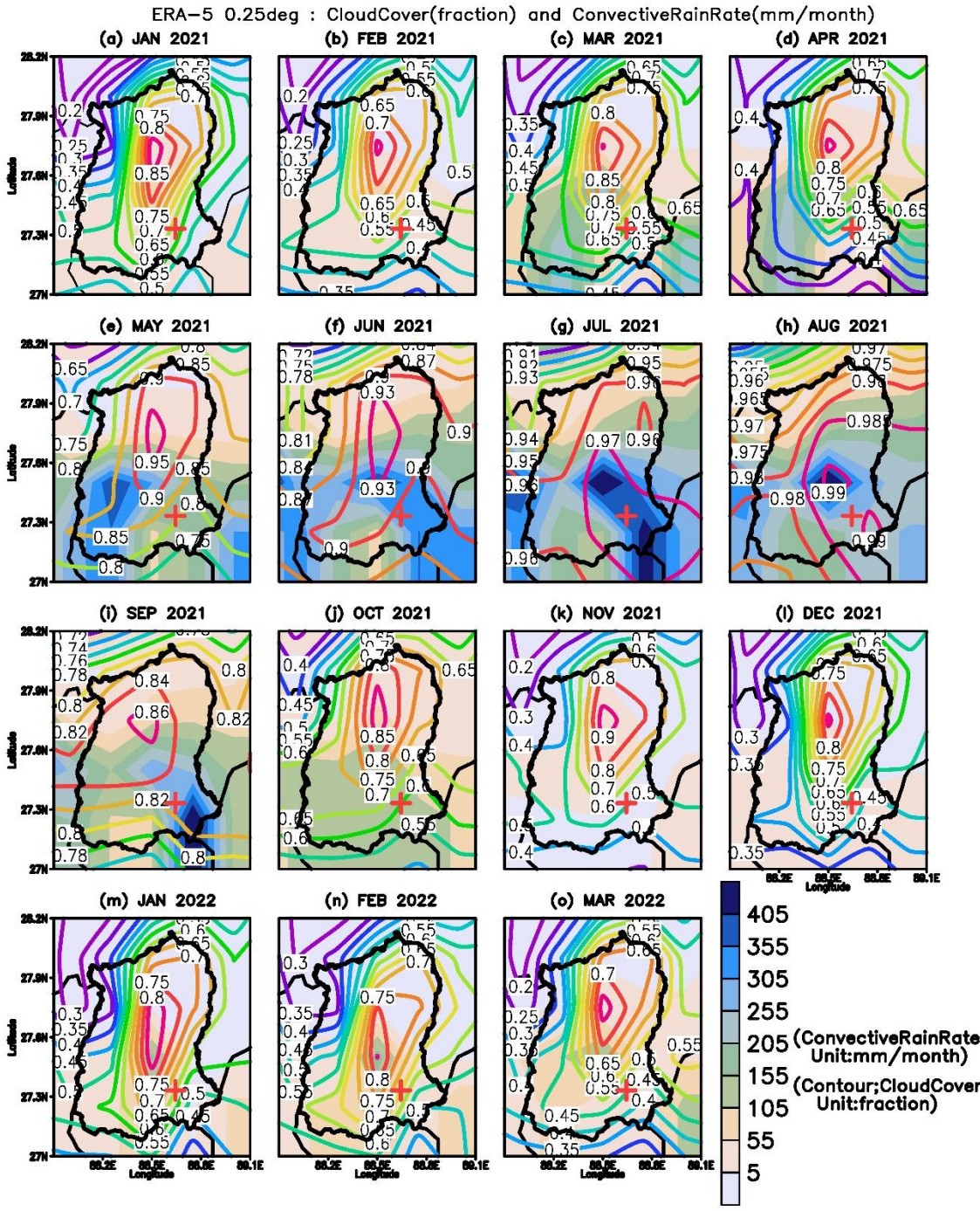

Figure 7. Monthly convective rain and total cloud cover during January 2021 to March 2022. The shading shows a convective rain pattern, and the contour shows a total cloud cover fraction. The (+) mark is a representation of the sampling location.

**List of Tables**

789     Table 1. The details of datasets used for the present study.

790

| Variables | Data sets | Years (Span) | Resolution | | Source | Reference |
|---|---|---|---|---|---|---|
| | | | Temporal | Horizontal | | |
| Black and Brown Carbon | Observation and analysis, data generated using Aethalometer AE33 | March 2021-March 2022 | Weekly | Point Location (Gangtok) | Original data generated | Present Study |
| Total precipitation | ERA5 (ECMWF) | 2021 to 2022 | Hourly | $0.25^O * 0.25^O$ | ECMWF https://cds.climate.copernicus.eu/cdsapp#!/dataset/reanalysis-era5-single-levels?tab=form | Hersbach et al., 2020 |
| Relative humidity | | | | | | |
| Temperature (2 meter) | | | | | | |
| Wind (surface wind) | | | | | | |
| Surface pressure | | | | | | |
| Dewpoint temperature | | | | | | |
| Net solar, and thermal radiation downward | | | | | | |
| LULC | LandSat-5, LandSat-8 and earth explorer USGS | December 2000, December 2010, December 2020 | 2000, 2010, 2020 | 30m, 30m | earth explorer USGS. https://earthexplorer.usgs.gov/ | earth explorer USGS. |
| LULC | Sentinel-2 Esri Inc. | December 2021 | 2021 | 10 m | Esri Inc. https://www.arcgis.com/home/item.html?id=d3da5dd386d140cf93fc9ecbf8da5e31 | Karra et. al., 2021 |

791