# Peer review of "1.0 Introduction"

_EGUsphere, 2023_

## Author Comment (AC1)

**Reply to the Comments:**

*The authors are thankful and appreciated to the reviewer and editor for their suggestions and insights. We are happy to incorporate all the suggests for better and comprehensive representation of the present work, and for making it easier to reader prospective.*

**The Comments**

This manuscript discussed about the Black carbon (BC) and Brown Carbon (BrC) variability based on measurements from March 2021 to March 2022 over Gangtok and their link with meteorological conditions obtained from satellite observations. They discuss the inter-relationship between BC and BrC emissions and their potential impact on climate, with the co-emission of CO2 and the impact on temperature and the potential role of BC/BrC as cloud condensation nuclei. I don't think the manuscript can be accepted as it is, the main reasons are addressed in the major comments.

| S. No. | Comments | Replies |
|---|---|---|
| *Major comments:* | | |
| 1. | I addressed only a small part of the grammatical issues in my minor comments, but the authors should seek the advice of an editor to revise the manuscript's language accuracy. | *Thank you for the suggestion we have addressed the issues and tried to rewrite the most part of the Manuscript.* |
| 2. | There is no mention of the mass absorption cross section used to estimate the mass concentration of BrC and considering the uncertainties around such value for BrC, it would be best to avoid using BrC mass concentrations and use BrC absorption coefficient at 370 nm instead. | *We have addressed the issues and incorporated the changes as per suggestion through entire Manuscript.* |
| 3. | Also, it is hard to see the link between LULC 2000, 2010 and 2020 and how it impacts the BC/BrC emissions and climate with only the measurement from 2021 and 2022. It might be easier to just state that the growing urbanization of the region may be at least partially responsible for the level of BC/BrC and CO2 observed. | *Yes, we agree with reviewer of putting It might be easier to just state that the growing urbanization of the region may be at least partially responsible for the level of BC/BrC and CO$_2$ observed.* *However, we put the actual scenario rather just telling, because there is not such study of LULC change to this region to cite. And just state that.* |
| 4. | There is a need for more references to support the different ideas mentioned in the discussion. | *We have tried to add some new relevant references in the discussion section as per suggestion.* *Shaddick et al., 2020; Rana et al., 2023; Huang et al., 2017; Stjern et al., 2023; Liu et al., 2020; Wang et al., 2020; Igarashi et al., 1988; Johnson and Hamilton, 1988;* |

| | | | *Sarkar, 2018; Liu et al., 2021; Wu et al., 2022; Yoo et al., 2014; Ohata et al., 2016; Ge et al., 2021; Wu et al., 2022; Zhu et al., 2021; Li et al., 2023a; Jung et al., 2023; Zhang et al., 2020; Li et al., 2023b; Davis, 2017; Chiodo et al., 2018; Xiao-lei et al., 2022.* |
|---|---|---|---|
| | | **Minor Comments:** | |
| 1. | | Page 4 line 125: what does the author mean by "fragile forest covers" and "The Gangtok is a densely populated city". | *The modification is made as mentioned here. (see page 4, line no. 128)* "Moreover, Sikkim has one of the most fragile forest covers. However, the Gangtok is densely populated city and capital of state Sikkim which is situated in the East Sikkim district (see figure 1a)." |
| 2. | | Page 6 equation 3: It is mentioned in the text "the negative log-log slope" so I would assume that a minus is missing in the actual equation. | *Thank you, yes it was mistakenly missed during editing, Now we made the correction. We have put in the proper way.* |
| 3. | | Section 3.1: The way all the equations are numbered and the reference to the supplement is very confusing. | *We made the correction and changed in the equation number throughout the entire manuscript. Please see section 3.1 and supplementary,* |
| 4. | | Page 6 Equation 1: The following writing is confusing "σBC + BrC(370 nm)" for the total absorption coefficient at 370, maybe write it $\sigma_{BC+BrC}$(370 nm). | *Yes, we agree to suggestion, and we made the change and corrected as per suggestion please see the equations 1,2,3 and 4.* |
| 5. | | Page 6 Equation 2: Again, the way the equation is written is confusing. "σBC($\lambda$) = $\beta$ $\lambda$−$AAE$BC" please rewrite the equation with "-AAE$_{BC}$" as an exponent to lambda. | *Yes, we agree to suggestion, and we made the change and corrected as per suggestion please see the equations 1,2,3 and 4.* |
| 6. | | Page 6 Equation 4: You mention in the text "Equation (3.16) was employed to determine σBrC (370 nm) by substituting σBC($\lambda$) at 370 nm, which was obtained using equation (3), into equation (3.13) (refer to supplementary methodology S1.1, S1.2, and figure 187 S2 for details). Shouldn't equation (4) be "$\sigma_{BrC}$ (370nm) = $\sigma_{BrC+BC}$ (370nm) - $\beta$(370nm)$^{-AAEBC}$"? | *we made the change and corrected as per suggestion please see the equations 1,2,3 and 4.* |
| 7. | | Page 8 lines 244-245: "BC BCbb, BC BCff" Please remove extra BC and "apparently". | *Thank you, we have removed the extra BC, and apparently. Please see page 8, line no. 248.* |
| 8. | | Page 8 lines 244-249: BCff peaks at 9am, CO2 at 10am and BrC/BCbb at 11am? What can you infer from | *We have looked into the suggestion and made the possible changes and rephased it with some addition, so that it does not* |

| | | |
|---|---|---|
| | these differences? Also, you mentioned that the same is observed for meteorological conditions? Can you be more precise because if their temporal variations were similar, it could mean that the meteorological conditions are driving the BC and BrC changes, which is not really the case here. | *contradict with itself. Please see page no 8, line 244-254.* |
| 9. | Page 9 lines 260-262: "BrC is found the highest with maximum fluctuation during 10th January to 30th March that is pointing towards winter wood burning for the subsistence as similar observed BCbb." Please rephrase | *Rephased the lines and made it easier for reader. Please see page no 9, line 265-268.* |
| 10. | Page 9 lines 262-267: What can you infer from the highest variations and concentrations of BC, BrC... and what could be the reasons behind such variation during March for BC and April for BrC ? | *The possible region has been added for the explanation. Please see page no 9, line 270-273.* |
| 11. | Page 9 line 270: "The good significant" and line 272 "strong significant correlation" remove significant in both cases. Furthermore, aren't BC and BCff and BCbb and BrC expected to correlate based on the way they are calculated? | *We rephased it as per suggestion and significant term is removed. Se page no. 9, line 277-280.* |
| 12. | Page 9 lines 273-275: "A good significant positive correlation between $CO_2$ and BC/BCff suggesting that fossil fuel burning is one of the causes of $CO_2$ concentration or vis versa." In figure 5, $CO_2$ doesn't seem correlated to anything else than himself. | *We have made the changes and rephrased the sentence along with the reference. Please se page 9, line 280-282.* |
| 13. | Page 9 lines 275-277: "Dewpoint temperature and $CO_2$ has strong significant positive correlation coefficient suggesting to positive radiative forcing of the $CO_2$." Some reference would be welcome and can the correlation be considered strong with a correlation coefficient of 0.22? | *The sentence is rephrased and some relevant references are also added. Please page 9, line 282-283.* |
| 14. | Page 9 lines 285-292: "However, cloud condensation nuclei formation and precipitation are prompted by | *We agreed with the reviewer and this is a probable explanation for comments. And explanation is also added to the* |

| | | aerosols (BC and BrC). Thereafter, BC and BrC have crucial role in precipitation mechanism." Also, BC being mainly hydrophobic, how good would BC particles be as CCN and which conditions would be required to efficiently play such role? | *Manuscript. Please see page 10, line 300-321.*

 *"However, BC particles can still act as CCN under certain conditions. For example, when BC particles mix with other aerosols, such as sulphates or nitrates, they can become more hydrophilic and more efficient as CCN (Moteki, 2023). Additionally, BC particles can be coated with organic material, such as brown carbon (BrC), which can increase their hygroscopicity and make them more efficient as CCN (Liu et la., 2020).*

 *The conditions required for BC particles to efficiently play the role of CCN depend on several factors, including their size, mixing state, and the atmospheric conditions. For example, smaller BC particles are more efficient as CCN than larger ones (Moteki, 2023). The mixing state of BC particles also plays a role, as externally mixed BC particles are less efficient as CCN than internally mixed ones (Liu et la., 2020). Atmospheric conditions such as relative humidity and temperature also affect the efficiency of BC particles as CCN. For example, higher relative humidity and lower temperatures can increase the efficiency of BC particles as CCN (Moteki, 2023).*

 *Moreover, BC particles are mainly hydrophobic and less efficient as CCN compared to more hydrophilic particles; they can still act as CCN under certain conditions. These conditions include the size and mixing state of the particles, as well as the atmospheric conditions such as relative humidity and temperature (Ohata et al., 2016; Moteki, 2023; Liu et al., 2020). Additionally, relative humidity over study region in very high during entire year with the favourable temperature. Thereafter, BC and BrC have crucial role in precipitation mechanism (Zhu et al., 2021; Li et al., 2023a)".* |
| 15. | Page 10 lines 293-299: Most phrases here are poorly written and need serious revisions to convey the observations clearly. | *Thank you, we rephased the lines see page 10-11, line 322-325.* |

| | | |
|---|---|---|
| 16. | Page 10 lines 298-299: How do you explain that the scavenging seem to be only affecting BCbb and not BC or BCff? | *Yes, it is affecting the all-constituents of BC, and BrC like $BC_{bb}$, $BC_{ff}$, etc. Please see page 10-11, line 320-321.* |
| 17. | Page 10 line 300: "pattern" instead of "patten" and what does the relative humidity and temperature justify? This sentence is not clear. | *The line is rephased and correction has been made. please see page 11, line 322-324.* |
| 18. | Page 10 line 308: "Figure 7 discusses" please rephrase. | *Rephased as per suggestion. Please see page 11, line 328-329.* |
| 19. | Page 10 lines 312-314: "approved" please use another verb and add reference regarding the important convective activity during the monsoon season in the Bay of Bengal. | *The sentence is rephased and references are added as per suggestion. Please see page 11, line 335-338.* |
| 20. | Page 10 lines 318-319: "supporting the convective rain (i.e., rain out scavenging) of all pollutants" do you mean scavenging of pollutant by convective rain here? | *Yes, Rephrased and Witten in the detail along with references. "The least concentration of BC, $BC_{ff}$, $BC_{bb}$, and BrC is observed during the monsoon months (Liu et al., 2020; Moteki, 2023). This observation supports the convective rain, as rain out scavenging, of all pollutants (Brooks et al., 2019; Liu et al., 2020; Moteki, 2023; Sankar et al., 2023). During the monsoon season, the region experiences high convective activity, which is added from the Bay of Bengal (Brooks et al., 2019; Liu et al., 2020; Moteki, 2023; Sankar et al., 2023). The convective rain is an effective process for removing air pollutants from the atmosphere (Liu et al., 2020; Moteki, 2023)."*
 *Please see page 11, line 341-350.* |
| 21. | Figure 2,3 and 4: Wouldn't box plot be a better option than average and standard deviation? If the blox plot are hard to read maybe had the median in the Table S2 and S3. | *Earlier we tried the box plot but it was not representing well so we put line with SD. We have added the median in the table S3.* |
| 22. | Figure 6: add the sampling site on, at least, one of the maps. | *The figure 6 and 7 are change and the point location of study site is pointed.* |
| *Supplementary information:* | | |
| 1. | Section 1.1: the notation is not consistent between the main text and the supplement (e.g. $\sigma BC(\lambda)$ and $b_{abs}(\lambda)$). | *The equations are changed. Please see supplementary as well as main manuscript section.* |
| 2. | Page 3: "ATN and BC relationship is given in figure (S7) for the daily | *Thank you, changed it.* |

| | | |
|---|---|---|
| | data." You are probably referring to figure S2 here. | |
| 3. | Page 3 Equation 3.4: instead of $B(\lambda)$, do you mean $BC(\lambda) = b_{abs}(\lambda) / \sigma_{abs}(\lambda)$ | *The equations are changed. Please see supplementary.* |
| 4. | Page 4 equation 3.7 and 3.8: Shouldn't "– aff" and "-abb" be exponent? | *The equations and numbering are changed. Please see supplementary.* |
| 5. | Page 5 equation 3.12: Do you mean BCff? | *The equations and numbering are changed. Please see supplementary.* |
| 6. | Page 6 Figure S2: Please correct the figure's caption. | *It is corrected as mentioned.* |
| 7. | Page 7 Figure S4: Please correct the figure "Total could cover" to "total cloud cover" | *It is corrected.* |
| 8. | Page 8 Figure S5: The BC and BrC data seem to still have zero did you estimate the limit of detection of the instrument? Should those points be included in the comparison? | *Yes, it is included. Zore is there because of two-digit values because the values were 0.00000x likewise.* |

---

## Author Comment (AC2)

**Reply to the Comments:**

*The authors are thankful and appreciated to the reviewer and editor for their suggestions and insights. We are happy to incorporate all the suggests for better and comprehensive representation of the present work, and for making it easier to reader prospective.*

**The Comments**

This study analyzed the seasonal and annual variation of black carbon (BC) and brown carbon (BrC) in Gangtok, Sikkim. Authors characterized the sources of BC, and discussed how meteorological conditions affected BC based on correlation analysis. Although the topic of this paper suits for EGUsphere, most results are basic and the discussion is not enough, leading to limited scientific information. In addition, the manuscript is poorly written and the language should be improved. Therefore, I do not think this manuscript meets the requirements of EGUsphere. The questions are listed below:

*Thank you for the suggestion we have addressed the issues and tried to rewrite the most part of the Manuscript.*

| S. No. | Comments | Replies |
|---|---|---|
| *Main comments:* | | |
| 1. | The authors use the ERA-5 reanalysis data for meteorological analysis. How do authors consider the uncertainties of the data set? | *Thank you for the suggestion we have addressed the issues and tried to rewrite the most part of the Manuscript.* *We have Discussed about the ERA5 uncertainties in the data section, Cited Some of research in the same region used ERA5 data for the meteorological study. Sharma et al, 2022, Kumar and Sharma, 2023.* *We have added AWS data along with ERA5 for support even through AWS data have huge discontinuity. But it can be seen that both data have almost similar pattern.* |
| 2. | Some results summarized in the abstract are not consistent with those analyzed in the paper. For instance, the authors mention when surface pressure is higher, the boundary layer is calmer, which results in the deposition of pollutants. In general, the deposition process leads to a decrease of pollutants. However, in the paper, the authors showed that higher surface pressure keeps the accumulation of pollutants, which is contradictory to the summary in the abstract. Please check these inconsistent contents. | *We have addressed the issues and incorporated the changes as per suggestion.* *The Sentence has been rephased. And also, explained more clearly this time.* *Also modified in the Abstract section.* |

| | | |
|---|---|---|
| 3. | The authors show many correlation efficiencies in the discussion section. Note that the correlation analysis indeed gives some evidence for what you observe, but they are not conclusive in this study. For example, the authors say 'The good significant correlation between BC and BCff suggested that the major contribution of the BC is fossil fuel burning'. The good correlation between BC and BCff does not necessarily mean that fossil fuel burning is the major contributor to BC. The authors should also give the proportion of BCff in BC to support this point. Please check other similar discussions in this section. | *Yes, we agree with reviewer. We have addressed the issue.*
*Thank you, we have addressed on the basis of table of the data set the actual contribution of BC, $BC_{ff}$, $BC_{bb}$, and BB%. And tried to described using correlation matrix for the same.*
*We also referred to the supplementary table S3 of monthly contribution of the BC, BrC, BCff, BCbb, and BB%.* |
| 4. | Lines 273-275, the authors conclude that fossil fuel burning results in the increase of CO2 based on the good correlation between BC and BCff, and think that if the increase of CO2 is not caused by fossil fuel burning, BC and BCff have poor correlation. Please prove this point. | *The modification is made as mentioned here. And some explanation after the correction is added to the Manuscript see in track-change mode, as well as accepted in page no. 9.* |
| 5. | Lines 287-292, please give the evidence that the decrease of surface pressure is caused by the vertical rising of air parcels. Authors mention that BC and BrC play important roles in cloud formation, please provide the evidence. | *The details discussion has been added to the discussion section with recent and relevant references. Please see page no. 10-11.* |
| colspan | *Minor Comments:* | |
| 1. | Line 177 and line 184, equations (3.15) and (3.16) are not contained in the supplementary information. | *The modification is made as mentioned here. (Please see page no. 6).* |
| 2. | Line 244, Table S11 is not found in the supplementary information. | *Thank you, yes it was typo mistake, now it is corrected.* |
| 3. | Line 248, it seems strange that the temperature increases during night time. Please explain why. | *We made the correction and changed in figure, the time was written wrongly, it must have started from 12PM. Please see the figure as well as discussion.* |
| 4. | Line 263, 'is' should be changed to 'are' | *Yes, we agree to suggestion, and changed.* |
| 5. | Line 270, delete 'good' or 'significant'. | *We have changed the sentence and rephrase as suggested, and remover the good. And some places we delated significant as per relevancy.* |

---

## Referee Report (RR1)

Authors have improved the language and shown more evidences to support their viewpoints. I think the manuscript has met the requirement of ACP after revised. However, some issues should be addressed before accepted.

Major

For the discussion of the effects of BC and BrC on CCN formation and precipitation, I still think that the conclusion is arbitrary, and the description dangers the manuscript. Authors insist that BC and BrC, no other aerosols, prompt CCN formation and precipitation in the study region. Yes, we believe that BC and BrC can act as CCN. However, the ability of BC and BrC to form CCN is not necessarily stronger than that of other aerosols with different compositions. In addition, authors mention that the study region is significantly affected by fossil fuel combustion and biomass burning. The primary aerosols (except for BC and BrC) and secondary aerosols (such as nitrate, sulfate) formed from their precursors might have stronger effects on CCN formation and precipitation. Although authors have added the discussions of the ability of BC and BrC to form CCN, the conclusion is not persuasive for not considering the effects of other aerosols.

Minor

Line 237 '. Which' change to ', which'

Line 285 and 288 delete 'strong' or 'significant'

---

## Author Response (AR2)

**Reply to the Comments:**

*The authors express gratitude to the reviewer and editor for their valuable suggestions and insights. We are pleased to incorporate all recommended changes to enhance the clarity and comprehensiveness of the present work, thus facilitating easier understanding for readers.*

**The Comments**

| S. No. | Comments | Replies |
|---|---|---|
| colspan | *egusphere-2023-702-referee-report-1* | |
| colspan | *Major comments:* | |
| 1 | For the discussion of the effects of BC and BrC on CCN formation and precipitation, I still think that the conclusion is arbitrary, and the description dangers the manuscript. | *Thank you for your thoughtful comments regarding the discussion of the effects of BC and BrC on CCN formation and precipitation in our manuscript. Your input is greatly appreciated, and we have carefully considered your concerns.* |
| 2 | Authors insist that BC and BrC, no other aerosols, prompt CCN formation and precipitation in the study region. Yes, we believe that BC and BrC can act as CCN. However, the ability of BC and BrC to form CCN is not necessarily stronger than that of other aerosol s with different compositions. | *We have addressed the issues and incorporated the changes as per suggestion. The discussion is added to the main manuscript. Please see line no 302-308, page no. 10, as highlighted in yellow colour.* |
| 3 | However, the ability of BC and BrC to form CCN is not necessarily stronger than that of other aerosols with different compositions in addition, authors mention that the study region is significantly affected by fossil fuel combustion and biomass burning. | *We have addressed the comments and added some point to the conclusion. Please see line no 396-401, page no. 13, as highlighted in yellow colour.* |
| 4 | The primary aerosols (except for BC and BrC) and secondary aerosols (such as nitrate, sulfate) formed from their precursors might have stronger effects on CCN formation and precipitation. Although authors have added the discussions of the ability of BC and BrC to form CCN, the conclusion is not persuasive for not considering the effects of other aerosols. | *We have addressed the comments and added some point to the conclusion as per suggestion. Please see line no 396-401, page no. 13, as highlighted in yellow colour.* |
| colspan | *Minor Comments:* | |
| 1 | Line 237 '. Which' change to ', which' | *The modification is made in manuscript as mentioned. Please, see as highlighted in line no. 239 in yellow colour.* |

| 2 | Line 285 and 288 delete 'strong' or 'significant' | *The modification is made in the manuscript as mentioned. Please, see as highlighted in yellow colour. (Line 287 and 290)* |
|---|---|---|

**egusphere-2023-702-referee-report-2**

**Minor Comments:**

| 1 | Page 5 line 124: "It is well well-established" removed the first well. | *We have made the change, please see the page no. 4, line no 113. Highlighted in the manuscript as yellow colour.* |
|---|---|---|
| 2 | Page 6 lines 154-161: "One of the main sources of uncertainty in using aerosol absorption measurements to estimate the BrC absorption coefficient at 370 nm BrC mass concentration is the fact that other species, such as black carbon and dust, can also contribute to the measured absorption. This can lead to overestimation of BrC mass concentration, particularly in environments where these species are also present. However, in the Sikkim region has one of the higher highest precipitation regions in the world and negligible contribution of to the dust pollution. Furthermore, there must be lesser over/under estimation. Therefore, the present study used mass concentration." The reviewer agrees with the first part of this statement about the dust not interfering with the absorption in the UV. On the other hand, the authors chose to report mass concentrations, but it is not mentioned how they pass from the BrC absorption coefficient at 370nm obtained in Eq. (4) to BrC mass concentrations. It would be nice to have at least a small line indicating the mass absorption coefficient (m2 g−1) used as such value can vary a lot based on compounds/sources/combustion process/fuels and lead to large uncertainties in BrC mass concentration estimation. | *The specific formula for the conversion is determined by the type of regression model utilized, such as linear regression in this case.*
 *For example,*

 *BrC concentration = m × Absorption coefficient + b,*
 *Where:*
 *'m' is the slope of the regression line (related to the sensitivity of the method).*
 *'b' is the y-intercept of the regression line.*
 *Absorption coefficient is the measured absorption coefficient at 370nm.* |
| 3 | Page 11 line 313: "A similar has been found for temperature", a word may be missing here. | *The modification is made in MS as per suggestion. Please, see as highlighted in yellow colour, page no. 9, line no. 286-287.* |
| 4 | Page 11 line 322: "A similar has been", a word may be missing here. | *The modification is made in MS as per suggestion. Please, see as highlighted in yellow colour, page no.10, line no. 295-297.* |

| 5 | Figure S5: The BC and BrC data seem to still have zero did you estimate the limit of detection of the instrument? Should those points be included in the comparison?

Yes, it is included. Zore is there because of two-digit values because the values were 0.00000x likewise.

Regarding this answer, the reviewer strongly suggests to have a look at the limit of detection of the instrument for BC, as 0.00000x seems fairly low. | *Yes, we agree to reviewer's suggestion, and we tried to address the comments, and corrected as per suggestion.*
*The Aethalometer AE33 is an aerosol instrument with a detection limit of <0.005 µg/m³ for a 1-hour period and a measuring range of 0.01 to 100 µg/m³. It has a programmable measuring frequency of 1 second or 1 minute and a programmable flow rate of 2 to 5 lpm.*
*Please see page no. 5, line no. 135-137. Please, see as highlighted in yellow colour.* |

---

## Author Response (AR3)

*We thankful to the unanimous reviewer for his/her valuable insight to the present work. We have addressed the suggestions and made all the changes as per suggestions. We also thank to the editor for his patience and help for making this work more enhanced and easier for reader.*

**Comments and Reply:**

Dear Authors,

I have received two reports from the reviewers. There are still some minor comments to be addressed as suggested by one of the reviewer. In addition, I would like to point out two issues that must be addressed in the revised version.

**Comment:**
1) Reporting BrC observations in mass concentrations are inappropriate unless the mass absorption coefficient (MAC) of BrC observed in the sampling locations are known. Otherwise, the BrC levels can only be reported as aerosol absorbance. Based on my understanding, the current manuscript does not include any determination/estimation/assumption of BrC MAC.

**Response:**

*We agree with the comments and we have addressed this in the supplementary and as well as main manuscript "*

$$BrC = \frac{b_{absBrC(370)}}{MAC_{BrC(370)}} \qquad\qquad Eq.\ (S13)$$

*The equation S13 calculates the Brown Carbon (BrC) mass concentration using the mass absorption coefficient (MACBrC) at a specific wavelength (370 nm) (Laskin et al., 2015). The MACBrC represents the ability of Brown Carbon to absorb light at that wavelength. MACBrC(370) is the mass absorption coefficient for Brown Carbon at 370 nm (Qin et al., 2018). The default value for MACBrC used is 4.5 m²/g.".*

*See line no. 116, page no. 5 of supplementary, and line no. 202, page no.7 in manuscript.*

**Comment:**
2) Although the positive statistical relationship between BC/BrC and thermal/solar radiation are observed, interpretating these relationships as direct evidence of positive radiative feedback caused by BC/BrC are inappropriate. I suggest to remove the related conclusion throughout the manuscript unless more comprehensive data analysis (e.g. modelling work) to support such conclusion.

**Response:**

*We agree with the comments and we have made the changes as per suggestion:*

*The statements are removed from the MS; line no 32 page 1, line no 361 page 12, line no 399 and page 13.*